# Thermomechanical Processing of a Near-α Ti Matrix Composite Reinforced by TiB_w_

**DOI:** 10.3390/ma13245751

**Published:** 2020-12-16

**Authors:** Hong Feng, Yonggang Sun, Yuzhou Lian, Shuzhi Zhang, Changjiang Zhang, Ying Xu, Peng Cao

**Affiliations:** 1College of Materials Science and Engineering, Taiyuan University of Technology, Taiyuan 030024, China; fenghong@tyut.edu.cn (H.F.); sunyonggang0153@link.tyut.edu.cn (Y.S.); lianyuzhou0093@link.tyut.edu.cn (Y.L.); zhangshuzhi@tyut.edu.cn (S.Z.); 2Department of Chemical and Materials Engineering, The University of Auckland, Private Bag 92019, Auckland 1142, New Zealand; sherry.xu@auckland.ac.nz

**Keywords:** titanium matrix composites, thermomechanical processing, mechanical properties, fracture toughness

## Abstract

To further improve the mechanical properties of the as-cast 7.5 vol.% TiB_w_/Ti–6Al–2.5Sn–4Zr–0.7Mo–0.3Si composite, multi-directional forging (MDF) and subsequent heat treatments were carried out to adjust TiB whiskers (TiB_w_) and matrix characteristics. The effect of various microstructures on the tensile properties and fracture toughness of the composites was analyzed in this paper. After MDF, the TiB_w_ are broken into short rods with a low aspect ratio and display a random distribution. Moreover, distinct microstructures were obtained after thermomechanical processing and different heat treatments. Both room-temperature and high-temperature tensile strength and ductility are improved after thermomechanical processing. By increasing the solution-treatment temperature, the microstructures transform from equiaxed to fully lamellar. A simultaneous improvement of the room-temperature and high-temperature properties is associated with the microstructural changes. In addition, the fracture toughness exhibits an increasing trend as the volume fraction of equiaxial α phases decreases. The lamellar microstructure demonstrates excellent fracture toughness due to deflection of the crack propagation path.

## 1. Introduction

Near-α titanium alloys are an important class of aerospace structural materials due to their lightweight and high strength at elevated temperatures. In the past few decades, several titanium alloys, e.g., IMI834, Ti-1100, BT36, and Ti60 series, have been developed for high-temperature applications [1,2]. However, as the service temperature is further increased, the high-temperature strength of these titanium alloys is challenging to meet their application requirements. The quest for titanium materials that can be used at even higher service temperatures has never ceased. Consequently, discontinuously reinforced titanium matrix composites (DRTMCs) have been proposed, which have high specific strength, specific modulus, and exceptional high-temperature durability [3,4,5]. Among the many reinforcements, one of the best is the TiB whisker that is fully compatible with the titanium matrix [6,7]. However, a common issue of TiB_w_/Ti composites is the less satisfactory mechanical properties, which is due to the non-uniform distribution of the reinforcement TiB_w_ and the coarse matrix microstructure [8]. Therefore, it is worth considering how to improve further the mechanical properties of as-cast TiB_w_/Ti composites [9].

Thermomechanical processing (TMP) has proven an effective way of modifying the microstructure and thus changing the mechanical properties of DRTMCs. A large number of studies have reported that traditional forging, extrusion, and rolling can significantly improve the properties of TMCs, but it is still challenging to produce fine grain and homogeneous materials [4,5,6]. Among many TMP methods, multi-directional forging (MDF) has proven effective in developing ultrafine-grained microstructures, weak texture, and a homogeneous distribution of reinforcement of DRTMCs. Zhang et al. [10] reported that after the forging of as-sintered TiB_w_/TA15 specimens, the ultimate tensile strength (UTS) and elongation at room temperature increased by 8.4% and 160%, respectively. In another study of 5 vol.% (TiB_w_ + Y_2_O_3_)/Ti composite [11], the UTS and elongation at room temperature were increased to 1.47 GPa and 7.2%, respectively, after the Ti composite samples were forged. Furthermore, the mechanical properties of DRTMCs can be further improved by subsequent heat treatments [12,13,14,15]. Li et al. [14] pointed out that after heat treatment, the tensile strength of (TiB + La_2_O_3_)/Ti composites increased, while the ductility decreased. The strength increment after heat treatments is attributed to the dispersed particles arising from the aging process. Huang et al. [16] investigated different heat-treating parameters of a 5 vol.%TiB_w_/Ti64 composite and found that an increased fraction of the transformed β phase leads to a significant improvement in strength at both room temperature and high temperatures. Hence, it is critical to investigate the microstructural evolution and associated strengthening mechanisms. In the aviation industry, toughness and strength are equally important for structural materials to avoid catastrophic fracture [17]. Fracture toughness is one important design criterion for obtaining a long-life safe flight [18]. Therefore, many investigations have been devoted to the fracture toughness and fracture mechanism of titanium alloys [19,20]. Jia et al. [18] investigated the influence of temperature on the fracture toughness and fracture mechanism of a Ti60 alloy and found that the intrinsic microstructural resistance, the tortuosity of the crack propagation path, and the crack tip plastic zone are the key factors that affect the fracture toughness of Ti60 alloy. He et al. [20] studied the influence of different microstructure characteristics on the fracture toughness of a BT-25 titanium alloy and found that the size and fraction of the lamellar α phases and the globularized α phases affect the crack propagation mode, which in turn affects the fracture toughness. Therefore, there is an inseparable relationship between the fracture toughness of titanium alloy and its microstructure characteristics. Although the intent of incorporating reinforcements to metals may be primarily to increase strength or wear resistance, these reinforcements do toughen the composites. In composite materials, crack bridging is the predominant toughening mechanism [17]. As for DRTMCs, little work has been undertaken to study the effects of the matrix microstructure and reinforcements on the fracture toughness.

In the present work, a near-α titanium matrix composite was subjected to MDF and subsequent high-temperature heat treatment to improve the mechanical properties. The main objective is to investigate the microstructural evolution during thermomechanical processing to reveal strengthening and toughening mechanisms.

## 2. Materials and Methods

This study deals with a Ti composite. The matrix is a near-α titanium alloy Ti–6Al–2.5Sn–4Zr–0.7Mo–0.3Si, while the reinforcement is 7.5 vol.% TiB whisker. Moreover, the composite was fabricated in an induced skull melting furnace (ISM), and sponge titanium (99.9%), high purity aluminum (99.99%), zirconium sponge (99.8%), Sn (99.99%), Al-Mo (50.5%), Si (99.7%), and TiB_2_ powder were raw materials. Meanwhile, TiB whiskers are synthesized in situ, as per the following reaction:
TiB_2_ + Ti = 2TiB.(1)

In order to ensure the compositional homogeneity, the melt was electromagnetically stirred for 5–10 min before casting. Then, the cast ingot was homogenized at 650 °C for 8 h, followed by furnace cooling.

Rectangular-shape billet with dimensions of 70 × 35 × 35 mm was cut from the as-cast ingot for MDF processing. Prior to forging, the β-transus temperature of the composite ingot was determined to be 1015 °C, using metallographic techniques. Subsequently, the ingot was subjected to MDF at 980 °C with a strain rate of 0.01 s^−1^, and the entire forging process was in a constant-temperature insulation environment, as shown in Figure 1a. These processes were described elsewhere [21]. In brief, the billet size is conserved after each pass. Only the loading axis varies by 90° after each pass, as shown in Figure 1b.

Specimens for heat treatment, fracture toughness tests, and tensile tests were cut from the as-forged billet by electric discharge machining. The heat treatment samples were encapsulated in evacuated quartz tubes to prevent oxidization before being loaded in a ZY-LS1600 furnace (ZYSYDL, Luoyang, China). Solution treatments were carried out at 975, 1000, and 1025 °C, respectively, followed by air cooling (AC). These temperatures were selected to represent a low-temperature band and high-temperature band in the α + β phase region, and β phase region. The solution treatment was followed by aging at 550 °C for 6 h and air cooling. The sample identification and respective heat treatments are summarized in Table 1.

The microstructures were observed by scanning electron microscopy (SEM) and electron backscatter diffraction (EBSD). The specimens were first ground with SiC grinding papers up to 2000 grit size and electrochemically polished in an electrolyte (60% methanol, 34% n-butanol, and 6% perchloric acid, in volume). Then, the polished specimens were etched with the Kroll’s reagent (10 mL HF, 30 mL HNO_3_, and 200 mL H_2_O).

Tensile and fracture toughness tests were conducted on the as-cast, as-forged, and heat-treated samples. The tensile samples with a cross-section of 4 × 2 mm and a gauge length of 18 mm were tested on an Instron 5500R machine (INSTRON, Boston, MA, USA) with a constant crosshead speed of 0.5 mm/min. For the fracture toughness tests, single edge notched bend (SENB) specimens were used, and the specimen size is shown in Figure 2. The ASTM E399 standard is used for the fracture toughness test. The movement speed of the indenter was 0.2 mm/min. The value of fracture toughness is calculated by Equations (2) and (3).
(2)$$K_{\mathrm{IC}}=\left(F_{\max}S/{\mathrm{BW}}^{3/2}\right)\times f\left(a/W\right),$$
(3)$$f\left(a/W\right)=3{\left(a/W\right)}^{1/2}\times \frac{1.99-\left(a/W\right)\left(1-a/W\right)\left[2.15-3.93\left(\frac{a}{W}\right)+2.70{\left(a/W\right)}^{2}\right]}{2\left(1+2a/W\right){\left(1-a/W\right)}^{3/2}},$$
where K_IC_ is the value of fracture toughness, F_max_ is the maximum load, S is specimen span, B and W are the thickness and width of the specimen, a is the notch depth, and f(a/W) is a geometric factor.

## 3. Results and Discussion

### 3.1. Microstructural Characterization

#### 3.1.1. Initial Microstructures

Figure 3 shows the microstructure of as-cast and as-forged TiB_w_/Ti composites. Figure 3a displays the basket-weave feature of an as-cast composite. The TiB appears to have a whisker-like morphology with a high aspect ratio (about 14.3). The matrix microstructure consists of α conolies and a lamellar α phase with a mean width of 5.8 μm. After MDF, the microstructure shows that the TiB whiskers were broken down into short rods with a low aspect ratio of 6.4, as presented in Figure 3b. These broken TiB rods are randomly distributed in the Ti matrix. Figure 4 is the EBSD analysis results of the as-forged composite, and the viewing surface of the microstructure is parallel to the loading direction (marked with the arrow CD in Figure 4b). It can be found that the Ti matrix features fine equiaxial α grains, along with residual β delineated at the α grain boundaries (Figure 4a,b). Meanwhile, the average grain size is quantified in Figure 4c, with a mean diameter of 4.6 μm. Nevertheless, a few coarse grains still exist in the matrix (marked with G1 and G2). Through further analysis of G1 and G2 grains, it can be observed that the coarse grains are split into several sub-grains by low angle grain boundaries. Moreover, Figure 4d is the misorientation profiles along arrows L1 and L2. It can be seen that the misorientation increased up to 11, which indicates that dislocations are highly active in G1 and G2 grains. It is believed that coarse grains (G1 and G2) will be broken into smaller equiaxed grains during further deformation.

#### 3.1.2. Microstructure Evolution after Heat Treatment

The microstructures of the composites after heat treatments are shown in Figure 5a–c. An equiaxed microstructure is observed in the sample if it is solutionized at 975 °C (HT1 in Table 1). When the solutionizing temperature increases to 1000 and 1025 °C, the overall microstructure changes into bimodal and fully lamellar structures, respectively. In the meantime, the size of the equiaxed primary α phase (α_p_) decreases with the solutionizing temperature. The α_p_→β phase transition occurs during the temperature ramping step, and the fraction of the β phase increases with temperature. In the following air cooling step, the majority of the β phase transforms into secondary α and some residual β phase, which are commonly known as the transformed β microstructure (β_T_). Finally, the matrix consists of different fractions of equiaxed α_p_ and β_T_.

Specifically, after HT1 heat treatment, the matrix consists of 93.8 vol.% equiaxial α_p_ and a small amount of β_T_ (Figure 5a). The mean size of equiaxial α_p_ measured is approximately 9.0 μm, which is larger than that of the as-forged counterpart. According to the lever rule, a lower solutionizing temperature corresponds to a smaller fraction of the α phase. When the solutionizing temperature increases to 1000 °C, the volume fraction of α_p_ decreases sharply to 11.5% and the mean size increases to 11.3 μm, while the width of the lamellar α phase is about 2.3 μm (Figure 5b). The fully lamellar microstructure with a width of 2.7 μm is observed in the samples solutionized at 1025 °C, as shown in Figure 5c. At this temperature, more α_p_ is converted into high-temperature β phase. It is worth pointing out that heat treatment does not change the morphology and size of TiB_w_, which is consistent with the previous study [16].

### 3.2. Tensile Properties

Table 2 shows the room-temperature tensile properties of the composites with different thermomechanical processing states. Firstly, the MDF has increased both yield strength (YS) and ultimate tensile strength (UTS) by 8.9% and 7.6%, respectively. In the meantime, the elongation to failure (δ) has dramatically increased (from 1.31% for the as-case to 3.97% for the as-forged sample). Secondly, heat treatments have further increased both YS and UTS, but at the expense of the elongation to failure. The strength increment is attributed to the gradual increase content of β_T_ with increasing of solutionizing temperature.

McEldowney et al. [15] found that different heat treatment processes influence the grain size, volume fraction, and morphology of the α phase and β phase. The increase of β_T_ structure after heat treatment will inevitably lead to the increase of hardness and strength of Ti64 matrix alloy. As shown in Table 2, it also found that the room temperature tensile strength of composites increases with solution temperature increasing. It is reported that the tensile strength of composites with fine lamellar microstructure is higher than those with equiaxial microstructure [22,23]. In this paper, the fine β_T_ increases with increasing solution temperature, thus resulting in the improvement of tensile strength. The elongation decreases with increasing tensile strength, and the as-forged specimen exhibits the best plastic obtained. In general, the plastic deformation of an equiaxed structure is superior to that of the lamellar structure. Compared to other microstructures, the equiaxed grains are less prone to stress concentration while more readily rotating to accommodate deformation.

After MDF, the original β grains are refined, and the number of grain boundaries increases, which increases the zones favorable for α grain nucleation, and TiB_w_ could also be used for heterogeneous nucleation to promote α grain nucleation. As a result, the grain size of the matrix alloy is significantly refined. So, the tensile strength and elongation both increased after thermomechanical processing treatment.

Table 3 lists the high-temperature tensile properties at 650 °C of the 7.5vol.% TiB_w_/Ti composite under different conditions. Thermomechanical processing can improve the high-temperature tensile properties of the composites in comparison to that of the as-cast sample. After MDF, the tensile strength, yield strength, and elongation of the composite are increased to 687.1 MPa, 612.3 MPa, and the elongation increased by 23%, respectively. The enhanced properties are a result of the microstructure being more homogeneous and the casting defects are being eliminated. At the same time, the fine equiaxed grains have high grain boundary migration ability, which is beneficial to coordinate deformation. The dislocation desnsity in the equiaxed α phase is high, and the dynamic softening phenomenon occurs during the high-temperature tensile test, resulting in its UTS being lower than that of the composite after heat treatment. Compared with the as-forged composites, the tensile strength of the composites after heat treatment is improved. After HT1, HT2, and HT3 treatments, the tensile strength increases to 734.9, 769.9, and 786.0 MPa, respectively. However, the elongation decreases to 16.13%, 16.82%, and 9.73%, respectively after HT1, HT2, and HT3 heat treatments. When compared with the equiaxed microstructure obtained after MDF, the HT2 and HT3 heat treatments changed the matrix microstructure to lamellar α. Consequently, the post-heat-treated samples have higher tensile strength but relatively lower elongation. The highest tensile strength was obtained in the sample subjected to HT3, and the matrix microstructure features fully fine lamellae. The fine lamellar microstructure plays a similar role to grain refinement, prevents the initiation of cracks, and improves the tensile strength. After heat treatment, the volume fraction of the fine lamellar α phase increases with the increase of the solution temperature; thus, the tensile properties are improved.

### 3.3. Fracture Toughness

Figure 6 shows the load–displacement curves from the fracture toughness test of the composites under different conditions. It can be seen that the trend of load variation with displacement can be divided into three stages: the initial straight elastic deformation stage, nonlinear plastic phase deformation, and final fracture failure. All samples exhibit a burst effect, which means that an instantaneous load drop occurs near the maximum load value (F_max_). The curve of the HT3 treated composite shows the maximum displacement indicating the highest fracture toughness.

The calculated fracture toughness values are shown in Table 4. After heat treatments, the fracture toughness increases, as compared with the as-forged sample. Meanwhile, the fracture toughness increases with solution temperature. There are two different types of toughening mechanism. One is the intrinsic toughening mechanism controlled by the inherent property of the material, whilst the other is the extrinsic mechanism depending on the size and geometry of the cracks, which can be directly reflected in the degree of tortuosity of the crack propagation path [24,25]. As the intrinsic resistance to fracture is related to the properties of the material itself, the composition of the composite material is the same, and the test parameters are consistent in this paper. Therefore, we just consider the extrinsic mechanism related to the tortuosity of the crack propagation. This extrinsic mechanism is mainly caused by microstructural feature difference, which will be further addressed in the following sections.

### 3.4. Fracture Mechanism Analysis

Figure 7 shows the fractograph of the 7.5 vol.% TiB_w_/Ti composites under different conditions. It is worth noting that the fracture morphologies among four microstructures are different. The fractograph of the as-forged sample (Figure 7a) consists of few dimples and tearing ridges, which is a quasi-cleavage feature. Numerous α cleavage planes are marked out with yellow circles in Figure 7a, indicating that the crack tends to propagate along these α grain cleavage planes. Thus, the cracks propagation resistance is relatively small, leading to a low value of K_IC_. Increased tearing ridges and dimples appear on the fractograph in the sample after the HT1 treatment, as shown in Figure 7b. It is noted that tearing ridges can change the crack propagation direction, leading to an increase in the tortuous crack propagation path and further improve the extrinsic resistance to fracture. In Figure 7c,d, there are some long dimples whose size is almost equal to that of lamellar α, and a few of them break in the middle. These observations indicate that the cracks may propagate along the lamellar α grain boundary or just cut through the lamellar microstructure. The secondary cracks (crack branching) are also observed in Figure 7d. Overall, the fractography of the sample with the fully lamellar microstructure is rougher than that of other samples.

As aforementioned, the tortuosity of the crack propagation is a significant factor to influence fracture toughness. Figure 8a shows the cross-section of the fracture surface and the crack propagation path of the as-forged composite. Relatively smooth crack propagation path is observed. It is clear that the cracks preferentially deflect along α grain boundaries, since the energy required to cut α grain is higher than that of bypassing them when cracks meet the fine equiaxed α grain during propagation. Meanwhile, it can be found that the secondary crack propagates along the edge of TiB_w_. It indicates that the fine equiaxed grain is favorable to the crack extension. A similar crack propagation process is found in Figure 8b; the crack mainly propagates along the coarse equiaxed α grain boundary. The reason why the K_IC_ value of the HT1 composite is higher than that of the as-forged composite can be summarized here. As the size of the α grains increases, the degree of deflection of the cracks increases, which makes the energy required for the cracks to cut through the α grains larger. The real crack propagation distance is greater as the degree of the crack deflection is greater, and the fracture toughness is greater [25]. Moreover, the crack deflection increases with the increasing of lamellar α phase volume, which causes a higher tortuosity of the crack propagation path, as shown in Figure 8c. Simultaneously, it can be observed that the crack propagation path is parallel to the TiB_w_ length direction. This is due to the difficulty of crack propagation through TiB_w_. Nevertheless, compared with the HT1 and HT2 composites, the crack extension of the HT3 composite becomes complicated. The crack’s propagation path follows three ways: parallel with lamellar α boundaries, cut through lamellar α, and bypass equiaxial α grain. Therefore, the coarse lamellar α microstructure causes the crack propagation path to be more tortuous (Figure 8d). It is noteworthy that the energy dissipated by the crack cutting α platelets is higher than that of crossing α colonies. The greater degree of the crack deflection, the more energy will be absorbed [17]. It is the main reason for the highest value of K_IC_ of the HT3 composite.

Figure 9 is the schematic diagrams of the crack propagation paths with different microstructures. The crack mainly propagates around the equiaxed α grain or along the lamellar α grain boundary instead of passing through it (Figure 9a–d). Moreover, in the deformation process, a crack tip plastic zone is formed at the intersection of the residual β phase and α phase. This results in local stress concentration, which makes the crack tend to propagate along the equiaxed α grain boundary. It is clear that the crack deflection increases during the transformation of the microstructure from full equiaxed to full lamellar in Figure 9. It is attributable to the larger crack tip plastic zone of the compact lamellar microstructure, which contributes to the higher fracture toughness [24]. In addition, due to the random geometric orientation of the α lamellae in the fully lamellar structure, it will lead to the generation of crack branches. The increase in the number of cracks can relax the strain field and absorb more energy. It can effectively improve the fracture toughness of the full lamellar structure material. It is noteworthy that the TiB_w_ can significantly inhibit the forward propagation of the crack and induce the crack to produce more branches (Figure 9b,c). Similarly, the micropores produced by TiB_w_ fracture can also induce secondary cracks (Figure 9b). Meanwhile, this also leads to the increase of crack deflection during its propagation. However, TiB_w_ can also cause cracks to propagate in a specific direction (along the edge of TiB_w_), which reduces the degree of crack deflection. It is not conducive to the fracture toughness. Therefore, we can draw a conclusion that the microstructure and grain size of the composites have a significant influence on the fracture toughness. It is an effective way to optimize the fracture toughness of the composites by inducing the crack to produce branching and making the crack propagation path more tortuous.

## 4. Conclusions

In the present study, we investigated multi-dimensional forging and subsequent heat treatments of a near α-Ti composite reinforced by 7.5 vol.% TiB_w_. The evolution of the microstructure and the improvement of mechanical properties are related to thermal working. The following main conclusions can be drawn from this paper.

MDF can significantly refine the grain of the matrix. Moreover, three distinct microstructures were obtained after heat treatment. With an increase in solution temperature, the volume of equiaxed α decreased and the size increased, respectively. Nevertheless, heat treatment can not alter the morphology of TiB whisker.

Room temperature tensile strength and elongation of 7.5 vol.% TiB_w_/Ti composite increase after thermomechanical processing. The tensile strength increases while elongation decreases with solution temperature. The highest tensile strength (1270.9 MPa) was obtained by a composite with a full fine lamellar microstructure after HT3.

The forged and heat-treated composites exhibit appreciably higher strength and ductility compared with as-cast composite at elevated temperature. The highest tensile strength (786.0 MPa) was obtained by the composite after HT3 with a fine whole lamellar structure. The fine lamellar structure is beneficial to the increase of strength.

The fracture toughness increases with increasing the solution temperature. The fracture mechanism is a quasi-cleavage fracture. The significant increase of fracture toughness can be attributed to the increasing deflection of the crack propagation path related to a decreased equiaxed α. The crack propagates along lamellar α boundaries, increasing the degree of tortuosity of the crack path.

## Figures and Tables

**Figure 1 materials-13-05751-f001:**
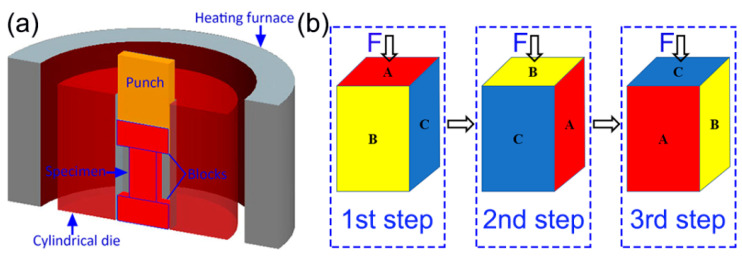
Schematic diagrams of the multi-directional forging (MDF) process: (**a**) equipment of MDF and (**b**) the sample processing routes during MDF.

**Figure 2 materials-13-05751-f002:**
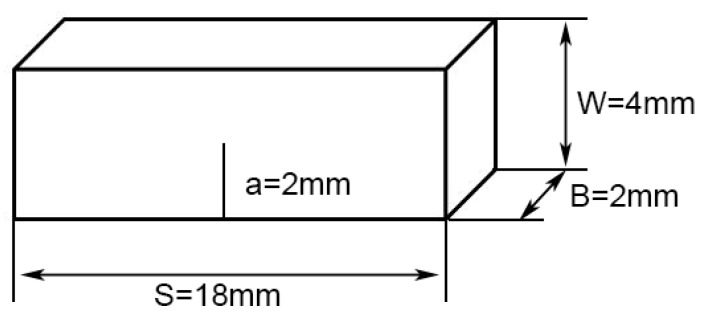
Single edge notched bend specimen used in fracture toughness tests.

**Figure 3 materials-13-05751-f003:**
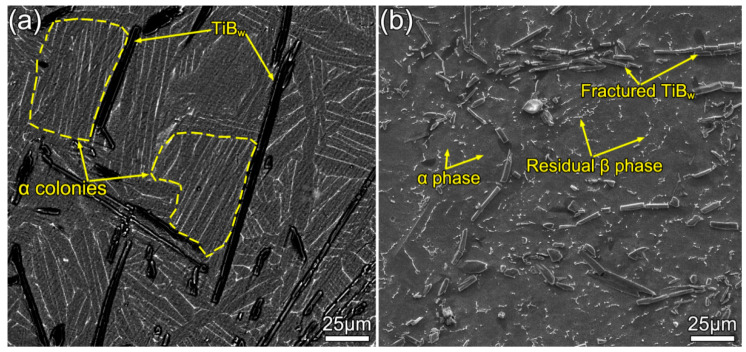
The microstructure of the TiB_w_/Ti composite: (**a**) as-cast composite and (**b**) as-forged composite.

**Figure 4 materials-13-05751-f004:**
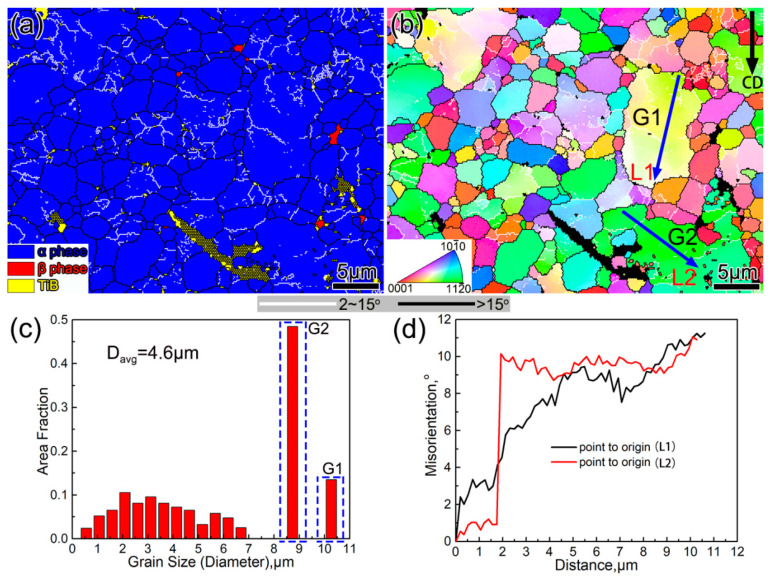
The electron backscatter diffraction (EBSD) analysis results of the as-forged composite: (**a**) the phase map; (**b**) the inverse pole figure map; (**c**) distribution of grain size; and (**d**) the misorientation profiles along the arrows L1 and L2, respectively.

**Figure 5 materials-13-05751-f005:**
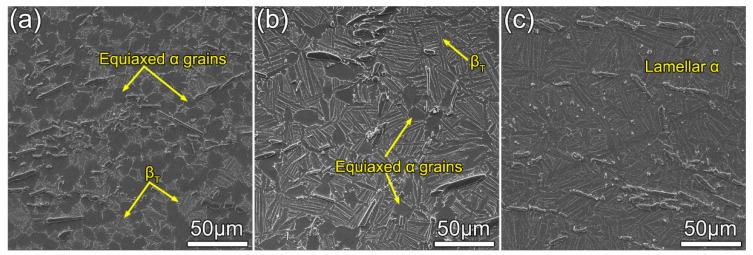
SEM micrographs of the Ti composites in various states: (**a**) HT1; (**b**) HT2; and (**c**) HT3. (The heat treatment conditions HT1–HT3 are listed in Table 1.)

**Figure 6 materials-13-05751-f006:**
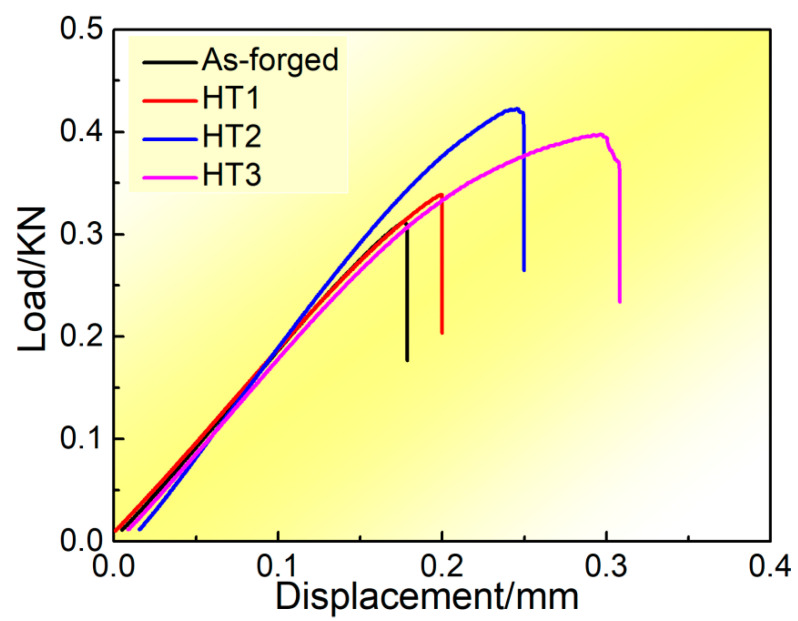
Load–displacement curves from the fracture toughness test of composites under different conditions.

**Figure 7 materials-13-05751-f007:**
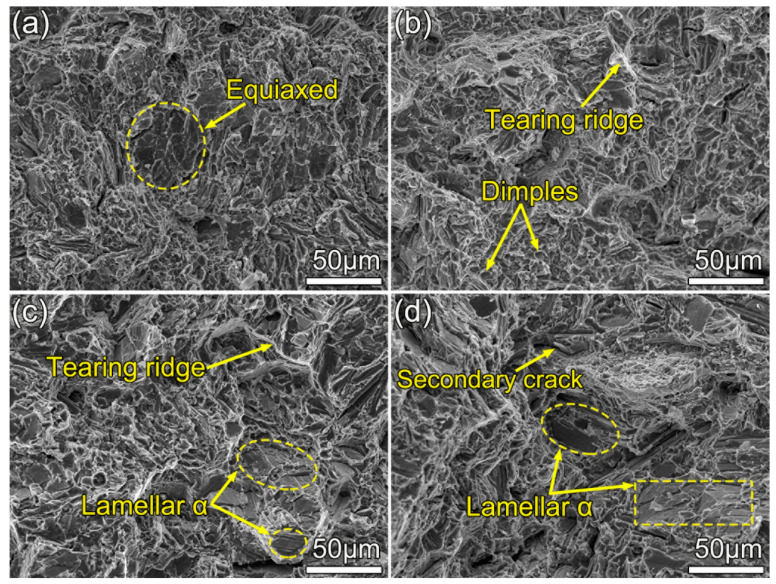
Representative fractographs of 7.5 vol.% TiB_w_/Ti composites under different conditions: (**a**) as-forged; (**b**) HT1; (**c**) HT2; and (**d**) HT3.

**Figure 8 materials-13-05751-f008:**
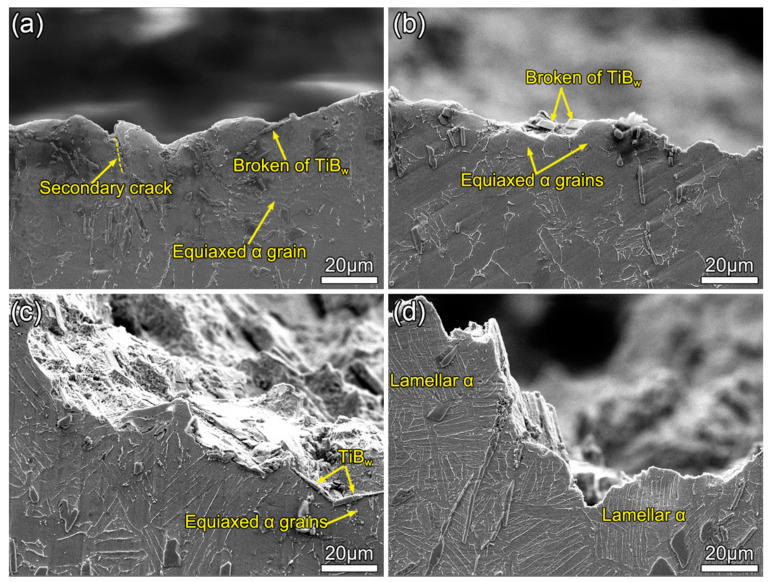
Micrographs of cross-sections of fracture surfaces: (**a**) as-forged; (**b**) HT1; (**c**) HT2; and (**d**) HT3.

**Figure 9 materials-13-05751-f009:**
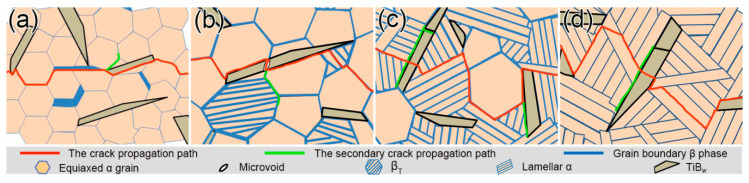
The schematic diagrams of the crack propagation paths in different microstructure features: (**a**) as-forged; (**b**) HT1; (**c**) HT2; and (**d**) HT3.

**Table 1 materials-13-05751-t001:** Heat treatment of the as-forged 7.5 vol. % TiB whiskers (TiB_w_)/Ti composite.

Sample ID	Solution Treatment	Aging Treatment
HT1	975 °C/0.5 h/AC	550 °C/6 h/AC
HT2	1000 °C/0.5 h/AC	550 °C/6 h/AC
HT3	1025 °C/0.5 h/AC	550 °C/6 h/AC

**Table 2 materials-13-05751-t002:** Room temperature tensile properties of the composites before and after thermomechanical processing.

Conditions	σ_0.2_ (MPa)	σ_b_ (MPa)	δ (%)
As-cast	1033.1 ± 7.5	1089.5 ± 8.2	1.31 ± 0.4
As-forged	1124.8 ± 9.4	1172.3 ± 10.1	3.97 ± 0.7
HT1	1143.2 ± 9.8	1197.7 ± 10.6	2.85 ± 0.6
HT2	1179.9 ± 10.2	1253.1 ± 12.7	2.48 ± 0.6
HT3	1201.6 ± 12.1	1270.9 ± 13.3	2.17 ± 0.5

**Table 3 materials-13-05751-t003:** High-temperature tensile properties of the composites before and after thermomechanical processing.

Conditions	σ_0.2_ (MPa)	σ_b_ (MPa)	δ (%)
As-cast	524.8 ± 2.7	606.0 ± 3.5	6.58 ± 1.2
As-forged	612.3 ± 3.2	687.1 ± 3.8	22.16 ± 3.4
HT1	650.4 ± 3.4	734.9 ± 4.5	16.13 ± 2.2
HT2	664.2 ± 3.8	769.9 ± 5.3	16.82 ± 2.4
HT3	707.9 ± 4.2	786.0 ± 6.0	9.73 ± 1.9

**Table 4 materials-13-05751-t004:** Fracture toughness (K_IC_) values for the 7.5 vol.% TiB_w_/Ti composites under different conditions.

Conditions	As-Forged	HT1	HT2	HT3
K_IC_ (MPa·m^1/2^)	47.1	52.0	54.9	58.5

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
