# Peer review of "Thermomechanical Processing of a Near-α Ti Matrix Composite Reinforced by TiBw"

_materials, 2020, doi:10.3390/ma13245751_

Round 1
Reviewer 1 Report
Very interesting work in which there are rather no factual and factual errors. My concerns concern only a few phrases and language.
1) I suggest rethinking and correcting the systematics of chapter 2. All issues are described in one sequence without any transitions or a description of the problem change.
2) Please add the potential use and reference for macro production of the material under test. I would also ask you to add a description of why this methodology of production was used, are they different, if so, how they relate to the MDF method.
3) Is it not a better solution to provide in Chapter 3 first the results of the tests of strength and fracture toughness, and then the structural analysis in order to explain the obtained dependencies?
4) The first sentence of the conclusions is probably not properly translated. Moreover, correct conclusions based on interesting results.
5) Required improvement of the legibility of markings in figures (especially 4, 6, 9).
Author Response
- I suggest rethinking and correcting the systematics of chapter 2. All issues are described in one sequence without any transitions or a description of the problem change.
Response: We have re-written this section. The modified parts at the manuscript are marked in red colour.
- Please add the potential use and reference for macro production of the material under test. I would also ask you to add a description of why this methodology of production was used, are they different, if so, how they relate to the MDF method.
Response: Thank you for your advice. We added this part of the description in the manuscript. The corresponding modifications are as follows:
Thermomechanical processing (TMP) has proven an effective way of modifying the microstructure and thus changing the mechanical properties of DRTMCs. A large number of studies have reported that traditional forging, extrusion and rolling can significantly improve the properties of TMCs, but it is still challenging to produce fine grain and homogeneous materials [4-6]. Among many TMP methods, MDF has proven effective in developing ultrafine-grained microstructures, weak texture and homogeneous distribution of reinforcement of DRTMCs.
- Is it not a better solution to provide in Chapter 3 first the results of the tests of strength and fracture toughness, and then the structural analysis in order to explain the obtained dependencies?
Response: we first presented the microstructure, followed by mechanical properties. Lastly, we presented the fracture analysis. we believe the flow of the presentation in this work is logic.
- The first sentence of the conclusions is probably not properly translated. Moreover, correct conclusions based on interesting results.
Response: Thank you for your advice. We have re-written this part. The corresponding modifications are as follows:
In the present study, we investigated multi-dimensional forging and subsequent heat treatments of a near α-Ti composite reinforced by 7.5 vol.% TiBw. The evolution of microstructure and the improvement of mechanical properties are related to thermal working. The following main conclusions can be drawn from this paper.
- Required improvement of the legibility of markings in figures (especially 4, 6, 9).
Response: Thank you for your advice. Higher-resolution figures (4, 6, and 9) have been added.
Reviewer 2 Report
The manuscript presents thermomechanical processing of a TiBw/Ti-6Al-2.5Sn-4Zr-0.7Mo-0.3Si composite to improve its mechanical properties. The authors carried out multi-directional forging and subsequent heat treatment to break TiB whiskers and to obtain fine lamellar microstructure. They conclude that tensile strength and fracture toughness were increased due to fine lamellar structure. This kind of mechanism has been already reported by many researchers but the specific information might be of interest to the readers of Materials. I would recommend it for acceptance after the minor points listed below are addressed.
- Page 2, line 88-89, the authors just wrote “These processes are described elsewhere [22]” but the forging conditions should be explained in the main body text so that the manuscript is self-contained.
- Page 3 Fig.1, any explanation of Fig. 1 cannot be found in main body text. Please remove Fig. 1 or add the explanation of it to the main body text.
- Page 3, line 109, the authors wrote “strain rate of 0.5 mm/min.” but I guess this should be crosshead speed, judging from the unit. In addition, the gauge length should be explained.
- Page 3, line 110, “Fig. 1” should be Fig. 2.
- Page 4, line 124, “Fig. 3a” should be Fig. 3b.
- Page 5, Fig. 4, the meaning of “CD” should be explained.
- Page 5, line 142, “Fig. 4a-c” should be Fig. 5a-c.
- Page 6, Table 2, were there yielding in stress-strain curves of these specimens? Isn’t this 0.2% proof stress?
Author Response
- Page 2, line 88-89, the authors just wrote “These processes are described elsewhere [22]”, but the forging conditions should be explained in the main body text so that the manuscript is self-contained.
Response: Thank you for your advice. We have added the description of Figure 1 and the relevant descriptions. See Section 2.
- Page 3 Fig.1, any explanation of Fig. 1 cannot be found in main body text. Please remove Fig. 1 or add the explanation of it to the main body text.
Response: Thank you for your advice. We have added the description of Figure 1. See Section 1.
- Page 3, line 109, the authors wrote “strain rate of 0.5 mm/min.” but I guess this should be crosshead speed, judging from the unit. In addition, the gauge length should be explained.
Response: Thank you so much for pointing out this. Yes, it was crosshead speed. Revision has been made.
- Page 3, line 110, “Fig. 1” should be Fig. 2.
Response: Thank you very much. We have had another thorough proofreading.
- Page 4, line 124, “Fig. 3a” should be Fig. 3b.
Response: Thank you.
- Page 5, Fig. 4, the meaning of “CD” should be explained.
Response: We added an explanation for arrow CD and the relevant modifications were made. See Section 3.1.1.
- Page 5, line 142, “Fig. 4a-c” should be Fig. 5a-c.
Response: Thank you.
- Page 6, Table 2, were there yielding in stress-strain curves of these specimens? Isn’t this 0.2% proof stress?
Response: There was no obvious yielding phenomenon in the stress-strain curves of these specimens. Therefore, the yield strength here is 0.2% proof stress, and the relevant modifications were made.
Reviewer 3 Report
Remarks
- Figure 3 - it is desirable to present the microstructure with the identical resolution
- The meaning of the sentence "The fracture surface contains quasi-cleavage feature" is not clear (line 319-320).
- It is need to correct the reference [8].
Author Response
- Figure 3 - it is desirable to present the microstructure with the identical resolution.
Response: Thank you for your advice. We have replaced the microstructure figure of as-cast composites.
- The meaning of the sentence "The fracture surface contains quasi-cleavage feature" is not clear (line 319-320).
Response: We changed the wording here into:
The fracture mechanism is a quasi-cleavage fracture.
- It is need to correct the reference [8].
Response: We have carefully revised the incorrect format in the references.